# A Surprisingly Effective Perimeter-based Loss for Medical Image Segmentation

**Rosana El Jurdi** [1,2]                          ROSANA.EL-JURDI@UNIV-ROUEN.FR
**Caroline Petitjean** [1]                         CAROLINE.PETITJEAN@UNIV-ROUEN.FR
**Paul Honeine** [1]                               PAUL.HONEINE@UNIV-ROUEN.FR
**Veronika Cheplygina** [4,5]                      V.CHEPLYGINA@TUE.NL
**Fahed Abdallah** [2,3]                           FAHED.ABDALLAH76@GMAIL.COM

[1] *Normandie Univ, INSA Rouen, UNIROUEN, UNIHAVRE, LITIS, Rouen, France*

[2] *Université Libanaise, Hadath, Beyrouth, Liban*

[3] *ICD, M2S, Université de technologie de Troyes, Troyes, France*

[4] *Computer Science Department, IT University of Copenhagen, Denmark*

[5] *Medical Image Analysis group, Eindhoven University of Technology, Eindhoven, The Netherlands*

## Abstract

Deep convolutional networks recently made many breakthroughs in medical image segmentation. Still, some anatomical artefacts may be observed in the segmentation results, with holes or inaccuracies near the object boundaries. To address these issues, loss functions that incorporate constraints, such as spatial information or prior knowledge, have been introduced. An example of such prior losses are the contour-based losses, which exploit distance maps to conduct point-by-point optimization between ground-truth and predicted contours. However, such losses may be computationally expensive or susceptible to trivial local solutions and vanishing gradient problems. Moreover, they depend on distance maps which tend to underestimate the contour-to-contour distances. We propose a novel loss constraint that optimizes the perimeter length of the segmented object relative to the ground-truth segmentation. The novelty lies in computing the perimeter with a soft approximation of the contour of the probability map via specialized non-trainable layers in the network. Moreover, we optimize the mean squared error between the predicted perimeter length and ground-truth perimeter length. This soft optimization of contour boundaries allows the network to take into consideration border irregularities within organs while still being efficient. Our experiments on three public datasets (spleen, hippocampus and cardiac structures) show that the proposed method outperforms state-of-the-art boundary losses for both single and multi-organ segmentation.

**Keywords:** Medical Image Segmentation, Convolutional Neural Networks, Prior-based Losses, perimeter length Constraint.

## 1. Introduction

Medical image segmentation consists of making per-pixel predictions in an image. The segmentation process is a key step in assisting early disease detection, diagnosis, monitoring treatment and follow up. Segmentation approaches based on convolutional neural networks (CNNs) are leading approaches in the field. However, the segmentation results they provide often suffer from anatomical errors, with holes, voids or high inaccuracies close to organ

boundaries (Bernard et al.). Recent works aim to alleviate this issue by integrating additional boundary or contour-based losses into the CNN (Kervadec et al., 2019a; Karimi and Salcudean, 2020; Caliva et al., 2019; Yang et al., 2019). In fact, (Kervadec et al., 2019a; Karimi and Salcudean, 2020; Caliva et al., 2019) show that contour-based losses could allow for more anatomical plausible segmentation when trained independently or in conjunction with a regional loss such as the soft Dice approximation or the cross-entropy. Contour-based losses often aim to minimize directly or indirectly the one-to-one correspondence between points on the predicted and label contour. Therefore, despite their significance, these losses are rather complex in nature and are characterized by hard gradients and high computational cost. Moreover, they often exploit distance maps to represent the change between predicted and ground-truth boundaries (Kervadec et al., 2019a) which in turn tend to underestimate the contour-to-contour distances given that the closest point is chosen systematically. As a result, the segmentation model may suffer from trivial local solutions (Kervadec et al., 2019a) or vanishing/exploding gradients. The problem becomes particularly challenging when the anatomical object under consideration has a complex shape with concavities or border irregularities.

In this work, a novel contour-based loss is proposed, which targets to constrain the perimeter or contour length of the organ to be segmented. Inspired by methods of (Shit et al., 2020), we extract contour maps from both the ground-truth and predicted segmentation maps. We then minimize the error between the predicted and ground-truth perimeter lengths by considering the sum over each respective contour via a mean squared error. We argue that by targeting the perimeter length rather than the point-by-point distance, the model will be able to take into consideration border irregularities, such as sudden corners or curvatures within organ shapes. In doing so, it avoids shrunken or expanded anomalies. Moreover, the simplicity of the proposed loss, being the mean squared error between two lengths, may play an important role in allowing the network to learn at a faster rate and with high efficiency.

The proposed loss is evaluated across three public datasets of different tasks and modalities. The spleen dataset is from the Medical Segmentation Decathlon and consists of CT images that target spleen segmentation. The ACDC dataset contains cardiac magnetic resonance images (MRI) and the goal is to segment the two ventricles and the myocardium. The Decathlon hippocampus dataset consists of segmenting two neighboring small structures in MRI images. These public datasets were chosen in such a way that the structures to be segmented are characterized by particular shapes and non-convexity as is shown in Figure 1. We test the significance of the proposed loss in both a single-organ segmentation setting and a multi-organ segmentation setting. Surprisingly, despite the simplicity of the proposed loss, it outperforms state-of-the-art contour losses for organs with non-convex shapes and maintains segmentation performance for simple shapes such as circles and holes. Moreover, the proposed loss allows for accurate delineation of common boundaries between neighboring organs in the multi-organ segmentation framework.

The rest of the paper is organized as follows. Section 2 provides a brief overview of the state-of-the-art contour-based losses. Section 3 elaborates on the proposed loss. Section 4 presents the datasets as well as the experimental settings. Section 5 analyzes model performance on the three datasets. Finally, Section 6 concludes with future works and perspectives.

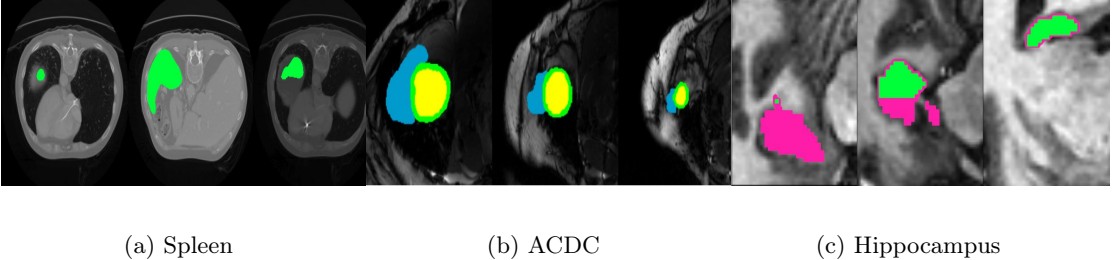

(a) Spleen        (b) ACDC        (c) Hippocampus

Figure 1: Sample images from the 3 datasets with ground-truth regions overlaid. (a) Spleen is in green, (b) right ventricle is in blue, left ventricle is yellow and myocardium in green, (c) brain hippocampus with outer (H1) and internal (H2) tissues in pink and green resp.

## 2. Related Work

In the literature, many research works have attempted to impose constraints at the level of the loss function in segmentation networks. One way to do so is to directly exploit the ground-truth map in order to enhance specific geometric properties, e.g. via distance map or Laplacian transform (El Jurdi et al., 2020; Bohlender et al., 2021). In this regard, two major contributions are the boundary loss, proposed by (Kervadec et al., 2019a), and the Hausdorff loss, proposed by (Karimi and Salcudean, 2020). Both works tackle the problem of contour optimization between ground-truth and predicted segments, to increase anatomical plausibility in their respective deep learning segmentation models. However, whereas (Karimi and Salcudean, 2020) conduct a direct point-by-point optimization of the predicted and ground-truth contours, (Kervadec et al., 2019a) derive, through (Boykov et al., 2001) graph theories, an equivalent term that fine-tunes the probability distribution via ground-truth distance maps. In this way, (Kervadec et al., 2019a) alleviate the high computational load demonstrated by (Karimi and Salcudean, 2020), resulting from the online computation of the predicted distance maps per each iteration and for all images in the dataset. Instead of weighting the probability distributions as in (Kervadec et al., 2019a), (Caliva et al., 2019) exploit distance maps as weighing factors for a cross-entropy loss term in order to improve extraction of shape bio-markers and enable the network to focus on hard-to-segment boundary regions. As a result, they give more weight to pixels lying in close proximity of the segmented anatomical objects than those that are far away. Instead of distance maps, (Yang et al., 2019) exploit Laplacian filters in order to develop a boundary enhanced loss term that invokes the network to generate strong responses around the boundary areas of organs while producing a zero response given pixels that are farther from the peripheries. In the same context, (Arif et al., 2018) extend the regular cross-entropy term with an average point to curve Euclidean distance factor between predicted and ground-truth contours in order to allow the network to take into consideration shape specifications of segmented structures.

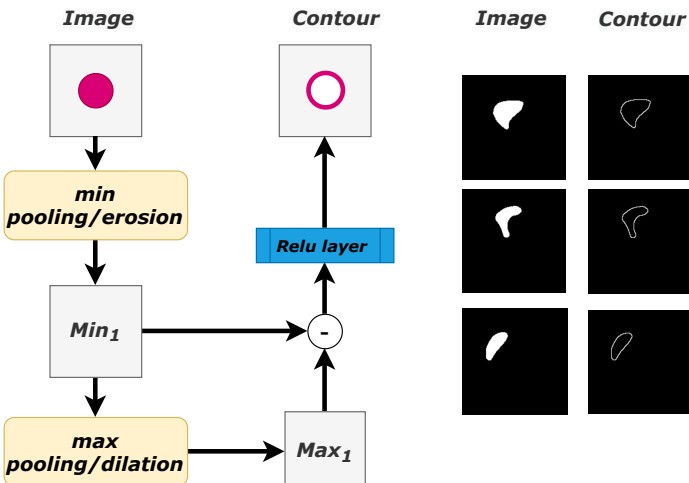

Figure 2: Principle of the contour function $F$: the difference (-) between the erosion (min-pool) and the dilation(max-pool) layers followed by a ReLu layer. Right side: example of ground-truth segmentation image and respective contour image.

## 3. Proposed Method

Many works in the field base their losses on distance maps in order to integrate geometric and location prior into the learning framework. Alternatively, we exploit contour maps produced via a combination of non-trainable max and min pooling layers.

### 3.1. Implementation of the contour function $F$

Implementation of the contour function $F$ is conducted via subtracting the erosion of the segmentation map from the dilation of the eroded map as shown in Figure 2. Dilation and erosion of the predicted and ground-truth maps are carried out via max and min pooling layers, followed by a ReLu layer. We note that $F$ can extract the contour of any image, whether it is a binary or a probability map. Hence, one can consider the contour function as a simple morphological gradient that can accommodate back-propagationin CNN training. The contour function $F$ is inspired by works of (Shit et al., 2020) that exploit this extraction strategy in order to integrate skeletonization constraints at the level of the loss function for tabular and vessel segmentation. (Shit et al., 2020) repeated this extraction process each time on the transformed image resulting from the previous iteration, consecutively summing over the output of all iterations in order to obtain the desired skeleton maps. After, they have taken into consideration the intersection over union of the precision and sensitivity between the ground-truth and respective skeleton maps. In our work, we have produced the contour maps by considering the subtraction of the erosion of the original segmentation map from the dilation of the eroded as is shown in Figure 2. After, we optimized the mean squared error between the sum of the ground-truth contour map representing ground-truth contour length vs. the predicted contour maps representing predicted contour length.

### 3.2. Loss Formulation

Let $\Omega \subset \mathbb{R}^2$ be the spatial image domain. Let $\mathbf{y}$ be the true label map and $\widehat{\mathbf{y}}$ the predicted label (probability) map, both of dimension $|\Omega|$. Consider $F$ to be a function that extracts the contour map of any image, as described in the previous section. The proposed loss is defined as a combination of the Dice loss (Milletari et al., 2016) and the perimeter-based loss weighted by $\lambda$ as follows:

$$\mathcal{L} = (1 - \lambda)\mathcal{L}_{Dice} + \lambda \mathcal{L}_{perim} \tag{1}$$

with

$$\mathcal{L}_{perim} = \left( \sum_{p \in \Omega} \widehat{y_p^F} - \sum_{p \in \Omega} y_p^F \right)^2 \tag{2}$$

where $y_p^F$ (resp. $\widehat{y_p^F}$) is the value of pixel $p$ in the map $F(\mathbf{y})$ (resp.. $F(\widehat{\mathbf{y}})$), equal to $y_p$ (resp.. $\widehat{y_p}$), if $p$ belongs to the contour, 0 otherwise. The contour function $F$ extracts for each of the predicted and ground-truth segmentation maps, a contour map of the segmented objects. The proposed loss then sums over the pixels for both the predicted and ground-truth contour maps and minimizes the mean squared error between them. Hence, one can consider the proposed perimeter-based loss as a regularizing term on the object perimeter.

## 4. Experiments

### 4.1. Datasets

In order to assess the added value of the proposed perimeter-based loss, several anatomical datasets are considered, with organs presenting varying characteristics in terms of size, shape, and border smoothness. The spleen dataset is a CT dataset from the Medical Segmentation Decathlon[1] whose objective is to segment a single organ (the spleen) characterized with a largely varying size and mild convexity issues at boundary levels, as is shown in Figure 1. The spleen dataset is composed of 41 patients divided into 32 for training and 9 for validation. The Hippocampus Dataset is also a medical Decathlon dataset designed for the segmentation of 2 neighboring tissues in the brain. It is composed of 263 mono-modal MRI scans divided into 206 patients for training and 56 for validation. The ACDC dataset is a cardiac cine MRI dataset consisting of 123 patients divided into 99 training and 24 validation. The task at hand is to segment simultaneously three elements of the heart: the left and right ventricular endocardium (LVC and RVC, resp.) and the myocardium (MYO). The segmentation task is rather challenging as the 3 components are in very close proximity to each other and are characterized by non-convex shapes or holes.

### 4.2. Implementation details

We deploy the unified U-Net based (Ronneberger et al., 2015) framework proposed in (Kervadec et al., 2019a,b, 2020) and modify the loss function accordingly. Training is done using a batch size of 8 and a learning rate of $10^{-3}$. The learning rate is halved if the validation

---

1. http://medicaldecathlon.com/

Table 1: Mean (± std) Dice index (%) and Hausdorff distance (mm) on the Spleen and RVC from ACDC dataset

| | Spleen Dataset | | ACDC Dataset (RVC) | |
|---|---|---|---|---|
| Loss | Dice index | Hausdorff | Dice index | Hausdorff |
| $\mathcal{L}_{Dice}$ | 76.80 ± 7.59 | 1.33 ± 0.28 | 81.22 ± 1.05 | 2.47 ± 0.04 |
| $\mathcal{L}_{perim}$ | 58.98 ±11.42 | 1.89 ±0.35 | 29.34 ±11.83 | 4.21 ±0.49 |
| $\mathcal{L}_{Dice} + \mathcal{L}_{Boundary}$ | 80.38 ± 5.46 | 1.34 ± 0.21 | 81.73 ± 0.81 | 2.35 ± 0.01 |
| $\mathcal{L}_{Dice} + \mathcal{L}_{HD}$ | 91.79 ± 2.67 | 0.92 ± 0.15 | 81.47 ± 1.01 | 2.42 ± 0.05 |
| $\mathcal{L}_{Dice} + \mathcal{L}_{perim}$ | **95.39 ± 1.26** | **0.71 ± 0.07** | **85.67 ± 0.50** | **2.21 ± 0.09** |

performance does not improve during 20 epochs. In the implementation of the contour function $\mathcal{F}$, max and min pooling were carried out via a kernel of size (3,3) and stride of 1. Since the contour extraction function is mainly composed of non-trainable layers, there is no considerable addition to the complexity of the network or the computational cost. Our code is publically available on GitHub [2]

The U-Net is trained with the loss as defined in Eq. 1, with a dynamic fine-tuning of the parameter $\lambda$ which was conducted in (Kervadec et al., 2019a). Thus, the parameter was set to 0.01 and increased by 0.01 per epoch for 200 epochs for the Spleen and ACDC datasets and for 45 epochs for the Hippocampus dataset.

For pre-processing, we have resized the images to a size of 256 × 256 and normalized them to the range [0, 1]. We have kept negative samples for training and validation. Each dataset was split into train and validation based on an 80 % , 20 % partition respectively and validated via three Monte-Carlo simulations (Arlot and Celisse, 2010).

## 5. Results and analysis

The proposed perimeter-based loss is evaluated in two segmentation settings: a single-organ segmentation setting, where we intend to train the network on the spleen dataset and on each of the structures of the heart in the ACDC dataset independently; and a multi-organ segmentation setting where simultaneous segmentation of the anatomical objects of the Hippocampus and ACDC dataset is carried out. We have compared the proposed loss $\mathcal{L}_{perim}$ to the Dice loss alone $\mathcal{L}_{Dice}$ and to two state-of-the-art contour-based losses: the Boundary loss $\mathcal{L}_{Boundary}$ (Kervadec et al., 2019a) and the Hausdorff loss $\mathcal{L}_{HD}$ (Karimi and Salcudean, 2020), each one trained in conjunction with the $\mathcal{L}_{Dice}$ given the same dynamic strategy presented in (Kervadec et al., 2019a) and explained in section 4.2.

### 5.1. Single Organ Segmentation

Results reported in Table 1 relative to spleen segmentation show that the proposed loss outperforms the Dice baseline and the state-of-the-art boundary losses by a large margin. When compared to the best state-of-the-art performance, the proposed perimeter-based loss has registered an increase by about 4% and a decrease in about 20% on the Dice index and Hausdorff distance respectively. Since the spleen is an organ characterized by a concave

---

2. https://github.com/rosanajurdi/Perimeter_loss

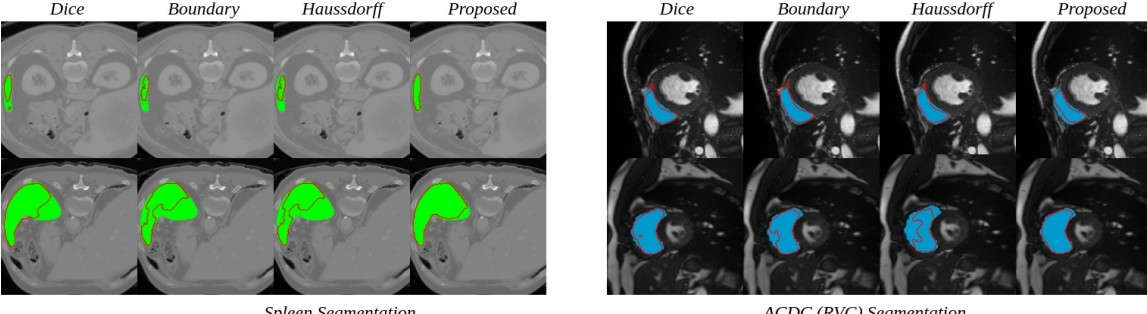

Figure 3: Segmentation results of the proposed loss against the Dice loss baseline and state-of-the-art losses in red with the ground-truth as a region filled with green for spleen and blue for the right ventricle (RVC) segmentation. Each row is a different image.

Table 2: Dice index and Hausdorff distance (pixels) results for ACDC (*simultaneous*) segmentation. RVC: right ventricular cavity, MYO: myocardium, LVC: left ventricular cavity

| Loss | Dice index | | | Hausdorff Distance | | |
|---|---|---|---|---|---|---|
| | RVC | MYO | LVC | RVC | MYO | LVC |
| $\mathcal{L}_{Dice}$ | 80.79 ±0.95 | 83.92 ±0.13 | 90.26 ±0.13 | 2.44 ±0.04 | 2.60 ±0.01 | 1.95 ±0.02 |
| $\mathcal{L}_{perim}$ | 23.19 ±13.38 | 26.53 ±9.63 | 9.78 ±4.07 | 4.29 ±0.17 | 4.59 ±0.43 | 4.39 ±0.05 |
| $\mathcal{L}_{Dice} + \mathcal{L}_{Boundary}$ | 81.04 ±0.87 | 84.16 ±0.83 | 89.53 ±0.74 | 2.41 ±0.05 | 2.57 ±0.01 | 1.95 ±0.02 |
| $\mathcal{L}_{Dice} + \mathcal{L}_{HD}$ | 80.54 ±1.30 | 83.91 ±0.85 | 88.98 ±0.90 | 2.33 ±0.04 | 2.65 ±0.01 | 1.98 ±0.01 |
| $\mathcal{L}_{Dice} + \mathcal{L}_{perim}$ | **84.49 ±0.57** | **86.22 ±0.41** | **90.69 ±0.41** | **2.19 ±0.03** | **2.55 ±0.04** | **1.94 ±0.02** |

border, we hypothesize that the significant decrease in Hausdorff distance highlights the ability of the proposed loss in accounting for varying curvature and border irregularities. This is illustrated qualitatively in Figure 3, where the proposed loss based on the object perimeter length and that is able to capture the specifications of the spleen contour.

Given that the RVC has a concave shape, it is similar to the spleen shape in many ways. Hence, we anticipate a similar behavior of the loss performance. Indeed, from Table 1, we gather that the proposed loss outperforms the best boundary state-of-the-art loss by 4% in Dice index and by more that 6% (from 2.35 to 2.21) in Hausdorff distances. Conducting further experiments where the other cardiac structures were segmented independently, we observe that the proposed loss maintains state-of-the-art performance when trained to segment organs with simpler shapes with circles (MYO) or holes (LVC). However, the results are not presented here due to the space limit.

## 5.2. Multi-organ segmentation

We have benchmarked the performances of the proposed loss on the Hippocampus dataset, which is composed of two neighboring structures, and on the ACDC datasets with the 3 cardiac structures. Multi-label segmentation performance on ACDC as shown in Table 2 reveals that training the model via the perimeter-based loss in conjunction with the Dice loss not only allowed improved segmentation on the irregular shaped RVC but also on the LVC

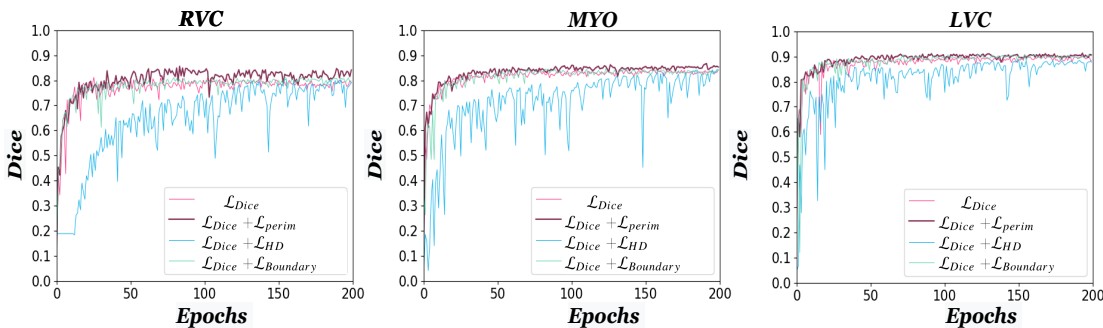

Figure 4: Curve evolution of Dice index on the three cardiac structures of the ACDC dataset in the multi-label segmentation. RVC/LVC: right/left ventricular cavity, MYO: myocardium

Table 3: Results for the Hippocampus Dataset. H1: green tissue, H2: pink tissue

| | Dice index | | Hausdorff Distance | |
|---|---|---|---|---|
| Loss | H1 | H2 | H1 | H2 |
| $\mathcal{L}_{Dice}$ | $49.37 \pm 1.76$ | $66.85 \pm 3.73$ | $3.89 \pm 0.14$ | $2.52 \pm 0.18$ |
| $\mathcal{L}_{perim}$ | $16.60 \pm 10.06$ | $36.21 \pm 1.68$ | $7.97 \pm 6.33$ | $3.10 \pm 0.12$ |
| $\mathcal{L}_{Dice} + \mathcal{L}_{Boundary}$ | $62.86 \pm 0.59$ | $75.52 \pm 0.48$ | $3.18 \pm 0.02$ | $2.16 \pm 0.02$ |
| $\mathcal{L}_{Dice} + \mathcal{L}_{HD}$ | $62.46 \pm 3.34$ | $74.12 \pm 3.42$ | $3.16 \pm 0.04$ | $2.44 \pm 0.25$ |
| $\mathcal{L}_{Dice} + \mathcal{L}_{perim}$ | $\mathbf{67.52 \pm 0.21}$ | $\mathbf{79.80 \pm 0.46}$ | $\mathbf{3.07 \pm 0.03}$ | $\mathbf{2.01 \pm 0.00}$ |

and MYO as well. This shows that improving segmentation performance on the hard-to-segment LVC has permitted proper delineation of other organs in its neighborhood. These observations were further validated by the evolution plot of the Dice index of its 3 structures as shown in Figure 4 given the 3 state-of-the-art losses against our proposed loss. Regarding the Hippocampus dataset, Table 3 shows that the proposed loss improves the Dice index and Hausdorff distance, for both tissues, by a considerable margin when compared relative to the state-of-the-art boundary losses. This verifies the ability of the proposed loss to properly delineate neighboring structures relative to other contour-based losses in the domain.

## 6. Conclusion

In this work, we propose a novel contour-based loss for medical image segmentation. The proposed loss considers the perimeter length of the segmented organ instead of exact boundary matching as is usually done in present state-of-the-art boundary losses in the field. We evaluate the proposed loss against three different datasets, which are characterized by varying and non-convex shapes. Despite the simplicity of the proposed loss, it is able to exceed state-of-the-art boundary loss performances by a large margin for both single and multi-organ segmentation. Future work includes customizing the proposed loss to accommodate multi-connected component organs and investigating different weighting strategies for the multi-organ segmentation setting under the proposed loss based on the degree of border irregularity of the considered organs.

## Acknowledgments

The authors would like to acknowledge the CNRS-Lebanon and AUF for granting a doctoral fellowship to R. El Jurdi, as well as the ANR (Project APi, grant ANR-18-CE23-0014) and the CRIANN for providing computational resources. This work is part of the DAISI project, co-financed by the European Union with the European Regional Development Fund (ERDF) and by the Normandy Region, and the WeSmile project funded by PHC VanGogh.

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
