# OpenReview forum: "A Surprisingly Effective Perimeter-based Loss for Medical Image Segmentation"
_MIDL.io/2021/Conference — MIDL 2021_

### Official Review · AnonReviewer1 · 2021-03-05

**Confidence:** 4
**Preliminary Rating:** 1
**Final Rating:** 3

**Summary:**

The authors propose a contour-based loss that targets to constrain the perimeter of the organ to be segmented.
They extract contour maps from both the ground-truth and predicted segmentation maps and then minimize the error between the sum over each respective contour via an L-2 norm. The proposed loss is evaluated across three public datasets.

**Strengths:**

This paper presents an exploration of a kind of contour-based loss with a simple strategy.
The new loss is validated on three datasets.
Results show that when it combines Dice loss, it outperforms Dice or other losses.

**Weaknesses:**

1. Some important details are missing or need clarification, for example, kernel size for min- and max- pooling,
2. Some necessary experiments are not presented. For example, the performance of using the new loss alone (without combining with Dice).
3. Experimental setting seems not reasonable, for example, tuning the model on the 'test' set.
4. Some errors were found in the table and figure. For example, the Hausdorff distance is smaller than one pixel which is not possible. The figure of segmentation results does not relate to their performance in the table.

**Deanonymize Review:**

no

**Detailed Comments:**

The authors present the motivation of the new loss and evaluate it on three datasets. However,  there are some concerns raised when I read through this paper:

1. In the introduction, the author made an argument - 'by targeting the perimeter rather than the point-by-point distance, the model will be able to take into consideration border irregularities, such as sudden corners or curvatures within organ shapes.' I like this argument, however, does this mean traditional boundary loss cannot deal with border irregularities? The author also mentioned non-convexity, but why their approach can deal with non-convexity and why others may not be able to deal with it is not clear in the paper.

2. In the method part, the author proposed to use min- and max- pooling to get the contour. What's the kernel size used? Does it mean the image size was reduced at some scale? And what would be the effect of such a down-sampling of the segmentation map?

3. In the method part, the scenario of the binary map is presented. Can this loss be used for multi-class as well? Can it be used in 3D segmentation? Please clarify this. I understand that the author presents the multi-organ segmentation but it's not clear that if they used a multi-class loss when training.

4.  The author proposed to combine the contour-based loss with Dice loss. The weight lambda seems not mentioned in the experiments.

5. The results of only using the new loss should be presented but missing in the table. Do the authors have a reason for this?


6. In Table 1, the Hausdorff distance is less than 1 pixel... Is this an error? In Figure 3, the result in the sub-figure for boundary loss seems very poor, is this a combination of Dice and boundary loss or is it the boundary loss alone? If it the former one, then it does not fit with the performances presented in Table 1.

7. In Section 4.2, the author mention 'the learning rate is halved if the validation
performance does not improve during 20 epochs,' however, there is no test set in the evaluation setting. This is not reasonable to me.


Minor:
The authors discuss several contour-based methods but I feel one important class of contour-based methods such as level set, snake model, and their recent advances [1] is missing.


References:
[1] Deep Snake for Real-Time Instance Segmentation



**Final Rating Justification:**

Most of my concerns were addressed.
Thus I change my rating to weak accept.

**Justification Of The Preliminary Rating:**

1. Related work needs more discussion on common contour-based methods, for example, the snake model and its recent advances.
2. Some important details are missing or need clarification, for example, kernel size for min- and max- pooling,
3. Some necessary experiments are not presented. For example, the performance of using the new loss alone (without combining with Dice).
4. Experimental setting seems not reasonable, for example, tuning the model on the 'test' set.
5. Some errors were found in the table and figure. For example, the Hausdorff distance is smaller than one pixel which is not possible. The figure of segmentation results does not relate to their performance in the table.

**Paper Type:**

both

**Questions To Address In The Rebuttal:**

Almost the points from the weakness part.

1. In the introduction, the author made an argument - 'by targeting the perimeter rather than the point-by-point distance, the model will be able to take into consideration border irregularities, such as sudden corners or curvatures within organ shapes.' I like this argument, however, does this mean traditional boundary loss cannot deal with border irregularities? The author also mentioned non-convexity, but why their approach can deal with non-convexity and why others may not be able to deal with it is not clear in the paper.

2. In the method part, the author proposed to use min- and max- pooling to get the contour. What's the kernel size used? Does it mean the image size was reduced at some scale? And what would be the effect of such a down-sampling of the segmentation map?

3. In the method part, the scenario of the binary map is presented. Can this loss be used for multi-class as well? Can it be used in 3D segmentation? Please clarify this. I understand that the author presents the multi-organ segmentation but it's not clear that if they used a multi-class loss when training.

4.  The author proposed to combine the contour-based loss with Dice loss. The weight lambda seems not mentioned in the experiments.

5. The results of only using the new loss should be presented but missing in the table. Do the authors have a reason for this?

6. In Table 1, the Hausdorff distance is less than 1 pixel... Is this an error? In Figure 3, the result in the sub-figure for boundary loss seems very poor, is this a combination of Dice and boundary loss or is it the boundary loss alone? If it the former one, then it does not fit with the performances presented in Table 1.

7. In Section 4.2, the author mention 'the learning rate is halved if the validation
performance does not improve during 20 epochs,' however, there is no test set in the evaluation setting. This is not reasonable to me.



**Special Issue:**

no

---

> ### Author Response · Authors · 2021-03-16
> **Response to AnonReviewer1**
>
> Weaknesses:
> 1. Hyper-parameter missing details: “The min and max pooling layers were carried out via kernel size (3,3) and a stride of 1”. We have added these specifications within the body of our paper.
> 2. Missing independent Experiments: Based on the reviewer's recommendation, we are conducting experiments on the perimeter loss on its own to check the possibility of learning and will include it in the modified version of the paper.
> 3. Experimental setting seems not reasonable: The model was not tuned on the test set. We have conducted 3 Monte carlo simulations where we benchmarked results on the validation set. The exact same framework was adopted in [kervadec el al. a (MIDL)], [kervadec el al. b (MIDL)], and [kervadec el al. 2020 (MIDL)] . We have replicated their framework and replaced their loss with our proposed loss, after which we have benchmarked the results.
> 4. Unit Error in table: We have changed the units to mm
>
> Questions to Address in the rebuttal:
>
> 1. In the introduction, the author made an argument - 'by targeting...:
>
> We don’t claim that regular boundary losses don’t have the ability to deal with border irregularities. However, we do admit to some advantages that the proposed loss has over the regular boundary-based losses: 1) The main essence of our work is taking advantage of the morphological gradient map that accommodates soft probabilities produced by network outputs. This morphological gradient map is much more representative of boundary variations than the regular distance maps that state-of-the-art boundary loss base on. 2) The proposed loss optimizes the error between predicted and ground-truth perimeter lengths. When extracting this feature from soft probabilities and optimizing it relative to its ground-truth label, this provides better results than just fine-tuning probability maps from network outputs via the distance maps. 3) Another point to take into consideration is the characteristics of the dataset and the effect of dataset characteristic on the validity and efficiency of the proposed loss. In this context, we refer the reviewer to our reply to reviewer 2 point-4 regarding the intuitive explanation of why the proposed method works.
>
>
> 2. Missing Hyper-paramter details:
>
> “The min and max pooling layers were carried out via kernel size (3,3) and a stride of 1”. We have added these specifications within the body of our paper.
>
> 3. In the method part, the scenario of the binary map is presented. Can this loss be used for multi-class as well? Can it be used in 3D segmentation?... :
>
> For multi-organ segmentation, we have computed the Mean Squared error for each organ and then took the mean between all the errors relative to each of the classes. Future work will include tweaking the loss with weighing maps according to the degree of border irregularity that the organ can have. For now, these are preliminary results verifying that perimeter-based regularization losses are indeed powerful tools that can improve segmentation performance in deep neural networks. For 3D segmentation, we may have some intuition on the external surface occupied by an organ but we don’t think that its implementation would be straight forward. We believe that a more complex loss that would capture the border irregularities may be more suitable in the 3D case.
>
> 4. The author proposed to combine the contour-based loss with Dice loss. The weight lambda seems not mentioned in the experiments:
>
> We refer the reviewer to our response to reviewer AnonReviewer4, Questions To Address In The Rebuttal, question 1 : “The exact same replica of the (Kervadec et al., 2019a) framework was exploited with only replacement of the boundary loss by the proposed loss. The term (alpha=1- lambda in our paper) was updated according to the experimental setting in (Kervadec et al., 2019a). Thus the dice was first given a weight of 1 and in each epoch 0.01 was stolen from the weighing factor on the dice loss and given to the regularization loss. We have mentioned this in the paper in section 4.2 : “The U-Net is trained with the loss as defined in Eq. 1, with a dynamic fine-tuning of the parameter λ. Thus, the parameter was set to 0.01 and increased by 0.01 per epoch for 200 epochs for the Spleen and ACDC datasets and for 45 epochs for the Hippocampus dataset.” However, we do apologize for improperly formulating equation 1) and equation 2) where  the 1- \lambda notation was not placed to finetune the L_{dice}. We have corrected this typing error in the body of our paper. “

---

> > ### Author Response · Authors · 2021-03-16
> > **Response to Reviewer1-Part 2**
> >
> > Questions to Address in the rebuttal:
> >
> > 5. The results of only using the new loss should be presented but missing in the table:
> >
> > The main idea why the results for training the proposed loss independently were not included was that our main objective was to show the role of the proposed perimeter-based loss in regularizing segmentation output. Intuitively speaking, the proposed loss mainly optimizes the over–all contour length of the organs. Hence, feature-wise it is not sufficient enough to allow proper learning and segmentation. The contour feature is not properly representative of the data. However, since the reviewers unanimously requested to see the loss performance independently of the dice , we have conducted training under just the contour loss and included the results in the tables.
> >
> >
> > 6. Table 1,Figure 3 issues:
> >
> > In table 1, we have corrected the Haussdorf unit to mm.  In figure 3, some examples of where the boundary loss did not perform well were included. OfCourse, the boundary loss performs well on other examples, but these are just some examples of the dataset where the boundary or Haussdorf loss does not function well whereas our proposed loss did. We agree this may not have been cleared in the paper and have modified the figure caption accordingly.  With regards to the boundary loss evolution, we don’t see how the results are poor. The lambda boundary + dice results resemble the dice loss at the beginning of the training since lambda is small after, it increments
> >
> > 7. In Section 4.2, the author mention 'the learning rate is halved if the validation performance does not improve during 20 epochs,' however, there is no test set in the evaluation setting. This is not reasonable to me:
> >
> > Training was done on the training set and validated on the validation set. Results in the table are on the validation set. Of course, having a test set would strengthen our claim. However, these are just preliminary results and presenting results on the validation set is an adopted method in many current papers. Examples include  (Kervadec et al., 2019a, MIDL), (Kervadec et al., 2019b, MIDL), and (Kervadec et al., 2020, MIDL). In fact, we have adopted the same experimental framework, training and testing strategies adopted by these papers, namely by (Kervadec et al., 2019a, MIDL).
> > Minor: The authors discuss several contour-based methods but I feel one important class of contour-based methods such as level set, snake model, and their recent advances [1] is missing:
> > We agree with the reviewer with regards to the significance of including and discussing the different contour extraction techniques. However, with the 8-page limits, we prioritized comparing relative the the reference losses we have compared experimentally to which already occupies the current permissible space.

---

### Official Review · AnonReviewer4 · 2021-03-06

**Confidence:** 4
**Preliminary Rating:** 2
**Recommendation:** Poster
**Final Rating:** 3

**Summary:**

The authors propose using the mean squared error between the prediction and target segmentation perimeter (sum of border calculated by dilation and  erosion layers) as a loss component added to Dice. This approach outperforms state-of-the-art options such as using only Dice loss, boundary loss or hausdorff loss.


**Strengths:**

The application of the max/min pool and ReLU to generate the borders is clever and, more importantly, traceable operations for feasibility in using perimeter as a loss.

Many experiments are presented including some very performant options such as Boundary Loss, in which surprisingly the perimeter approach led to better segmentation metrics in three very different medical imaging datasets. A highlight also has to be made to the use of a unified UNet framework, being the same used by Boundary Loss’s authors.

The paper is very well written and formatted.


**Weaknesses:**


There are two aspects not included in this study that would improve my recommendations for this submission, although i understand there might not be enough time now to include them:

1) Other loss functions could have been included in the comparison, such as cross entropy or generalized dice loss*.

2) Evaluation of perimeter loss’s performance on a highly unbalanced dataset, e.g: with a large percentage of background labels. Segmentation of the hippocampus would have been an example of that if the MSD dataset wasn’t already cropped to the hippocampus area.

In theory, it worries me that the perimeter does not give a "direction" for optimization. However this is partially adressed by the authors when stating that the perimeter is acting as a regularization of Dice.

Finally, there is a major concern that will refrain me from recommending for acception of this submission until addressed. Since the authors claim to have a loss that overperforms Boundary Loss, Dice, and HD, open source code with implementation of your proposal and how you implemented the other losses used in the comparisons is necessary. The reason for this is that in my experience of using Boundary Loss, its initial behaviour is very similar to Dice Loss, due to the high weight given to Dice in the beginning of training. This is not what i see in Figure 4, with the initial behaviour very different from Dice Loss, and raises concerns if Boundary Loss was implemented correctly.


-------------------------------------------------------------------------------------------------------------------------------------------------------------------------

* Sudre, Carole H., et al. "Generalised dice overlap as a deep learning loss function for highly unbalanced segmentations." Deep learning in medical image analysis and multimodal learning for clinical decision support. Springer, Cham, 2017. 240-248.


**Deanonymize Review:**

no

**Detailed Comments:**

Figure 4: What is the x axis of the plots? Epochs? Also i would recommend fixing all the graphs y range (e.g: from 0 to 1.0 or 0.9). Another suggestion is making it clear in the graph that the y axis value is Dice, even though it is clear on the caption.

In some parts of the paper, Boundary Loss refers to Dice+ LBoundary (Figure 4), in others, only to LBoundary (Section 4).

The quality of Figure 2 could be improved, try to export in a vectorized format (.eps, .pdf...).


**Final Rating Justification:**

The authors have provided code and much needed clarifications and changes. After the rebuttal and the changes to the manuscript the authors mentioned, I feel this work should be accepted as a poster.

**Justification Of The Preliminary Rating:**

The paper is very well written and presents an interesting idea that could be easily implemented in other works to possibly improve segmentation performance. However, I would need answers to the points I raised in this review to provide an acceptance.


**Paper Type:**

both

**Questions To Address In The Rebuttal:**

I am open to debate any point risen in the previous sections.

In your implementation of LDice + LBoundary, did you use the original Boundary Loss implementation? Specifically, did you use the shifting weight term alpha?

The weight for Dice in your proposed loss is always 1? Have you tested using a shifting weight (which decreases for dice and increases for the perimeter)?



**Special Issue:**

no

---

> ### Author Response · Authors · 2021-03-16
> **Official Response to AnonReviewer4**
>
> We thank the reviewer for their input. We summarize the reviewers points +discussions as follows:
> - Weaknesses: Other loss functions could have been included in the comparison, such as cross entropy or generalized dice loss:
>
> We agree on the interest of such comparison. Other loss functions such as the generalized dice, the cross entropy, the size loss proposed by (Kervadec et al., 2019b), and the clDice loss proposed by (Shit et al., 2020) as well will be added and compared relative to the proposed loss in future extensions to the paper.
>
> - Weaknesses: Evaluation of perimeter loss’s performance on a highly unbalanced dataset:
>
> We are planning on running the proposed loss on a variety of datasets of different characteristics if our paper gets extended to a longer version, in order to pin-point its strengths and limitations across the different characteristics. However, with the 8-page limit of this conference, it is not very feasible.
>
> - Weaknesses : Reproducibility and Code release:
>
> The code is now released on Github via this link: https://github.com/rosanajurdi/Perimeter_loss. To ensure reproducibility, we have used the exact framework as the papers of (Kervadec et al., 2019a) and have inserted the proposed loss as well as the Hausdorf floss respectively.  The Hausdorff loss function is provided in this Github code: https://github.com/JunMa11/SegLoss/tree/master/losses_pytorch
> We believe that this experimental setting is correct given the fact that it is adopted by 3 works published in the MIDL conference.
>
> - Weaknesses : Behavior of the Boundary loss:
>
> Indeed, the boundary loss (L_{dice} + \lambda L_{boundary} does mimic the L_{dice} performance at the beginning of the training. This is verified in fig 4 when comparing the green curves (Boundary + dice) to the pink (L_{Dice}= curves. This is true across all ACDC structures minding the performance drop which is resulting from the change in the weight lambda.
>
> - Detailed Comments:
>
> Based on the reviewers suggestions, we have modified the visual issues in the paper with regards to Fig 4. In all the experiments, we have conducted training of the boundary loss in conjunction with the Dice loss. This is verified by the evolution of the performances under the compared losses in fig 4. We did not compare relative to training via just the boundary loss. We agree this may have not been explicitly pointed out in the paper so we have modified the body of our paper to accommodate this clarification.
>
> - Questions To Address In The Rebuttal:
> 1. In your implementation of LDice + LBoundary, did you use the original Boundary Loss implementation? Specifically, did you use the shifting weight term alpha?
>
> Yes, the exact same replica of the (Kervadec et al., 2019a) framework was exploited with only replacement of the boundary loss by the proposed loss. The term (alpha=1- lambda in our paper) was updated according to the experimental setting in (Kervadec et al., 2019a). Thus the dice was first given a weight of 1 and in each epoch 0.01 was stolen from the weighing factor on the dice loss and given to the regularization loss. We have mentioned this in the paper in section 4.2 : “The U-Net is trained with the loss as defined in Eq. 1, with a dynamic fine-tuning of the parameter λ. Thus, the parameter was set to 0.01 and increased by 0.01 per epoch for 200 epochs for the Spleen and ACDC datasets and for 45 epochs for the Hippocampus dataset.” However, we do apologize for improperly formulating equation 1) and equation 2) where  the 1- \lambda notation was not placed to finetune the L_{dice}. We have corrected this typing error in the body of our paper.
>
>
> 2. The weight for Dice in your proposed loss is always 1? Have you tested using a shifting weight (which decreases for dice and increases for the perimeter)?
>
> The experiments we have conducted are tested via this dynamic training strategy (shifting weight strategy) the exact same way as the experiments of (kervadec et al a.). We apologize for the error in formulating equation (1). We have corrected the equation to be in accordance with the dynamic update strategy proposed by  (kervadec et al).

---

### Official Review · AnonReviewer2 · 2021-03-09

**Confidence:** 5
**Preliminary Rating:** 3
**Recommendation:** Poster
**Final Rating:** 3

**Summary:**

This paper is about improving the quality of the predicted segmentation, by regularizing its boundary.

Contrary to other methods such as Boundary or Hausdorff losses (that attempt an exact boundary matching), the authors try to minimize the difference between the perimeter of the predicted segmentation, and the perimeter of the ground truth. The effect "smooth" the predicted boundary, by penalizing sparse predictions.

Perimeter minimization is common in traditional computer vision, but is subject to a shrinking bias. As here we have a label to compare it to, this is not a problem. The perimeter is computed with a tiny non-trainable CNN module, that is compatible with continuous probabilities (this is important for back-propagation).

The experiments are done on several datasets, including some multi-class settings.

**Strengths:**

- Simple method (a few standard non-trainable CNN layers)
- Good reminder of the usefulness of perimeter-based regularizers
- Thorough evaluation (three datasets, two metrics)
- Good and convincing results

**Weaknesses:**

- More details on the perimeter module would be useful, and providing an example with an input made of continuous probabilities would help
- Discussion on the computational cost of the perimeter module is missing (I think it will be negligible, but numbers are required)
- Discussion on the limitations in 3D (or any setting requiring sub-patching) is missing. This isn't an issue here, but it is worth highlighting. For context, cross-entropy losses and derivative can be sub-patched, dices et al. no. Boundary loss can, Hausdorff loss cannot.

**Deanonymize Review:**

no

**Detailed Comments:**

I think the authors should focus more on the regularizer aspect of the proposed loss. I consider that the main contribution is the **reminder** that perimeter-based regularizers can be powerful in the deep learning setting. There is then different ways to implement the perimeter computation.

This is where the paper should be improved, as there are a few different ways to compute the perimeter (which is, at the end of the day, is the sum of the spatial gradients $\frac{\partial s_\theta}{\partial x\partial y}$):
- what the authors proposed;
- by hand, shifting the prediction by one pixel on each axises, and computing the difference (this is a bit crude but it works decently);
- this is also doable with a simple convolution pass;
- [1] did something related, by training a module that learns to compute the boundary (probably overkill in my opinion, but that should be discussed);
- standard method involving the image Laplacian (perimeter =$s_\theta^\top L s_\theta$, where $s_\theta$ is the flattened tensor of predicted probabilities and $L$ is the image graph Laplacian).

The authors do not discuss those different options, that all comes with different tradeoffs and different "sensitivity" to sparse predictions. I'd like to see a benchmarking of all methods on "real" examples (without necessarily training a network on it, simply testing a few representative inputs and displaying the result).

Perimeter minimization, as a regularizer, is quite common in traditionnal computer vision, I think it should be discussed more in the related works section (GraphCut, total variation, and other methods based on Potts models ultimately balance the length of the perimeter and the unary potentials---which is what you are attempting).

Minor:
- In the introduction, "the segmentation model may suffer from local solutions" -> modify to "trivial local solutions"
- Figure 4 font size should be increased, the colors possibly tuned (my color printer really struggled with them)
- (Karimi et al, 2019) is actually a TMI, not an arxiv anymore

---
[1] Moltz, J. H., Hänsch, A., Lassen-Schmidt, B., Haas, B., Genghi, A., Schreier, J., ... & Klein, J. (2020, April). Learning a loss function for segmentation: a feasibility study. In 2020 IEEE 17th International Symposium on Biomedical Imaging (ISBI) (pp. 357-360). IEEE.

**Final Rating Justification:**

I am happy with the answers that the authors provided to me and the other reviewers.

**Justification Of The Preliminary Rating:**

As the results are already convincing and sufficient for the conference (plus, with the 8 pages limit, there is not much space to add more content), I am mostly discussing for a future journal extension (recommending for the special issue).

**However**, the conference version still needs to clarify the design choice to compute the perimeter, and to discuss more the regularizer property of their loss. The opposition to the boundary losses is interesting, but slightly irrelevant as they do not work on the same objective (contour smoothing vs exact boundary matching). The notation should also be improved.


**Paper Type:**

both

**Questions To Address In The Rebuttal:**

See weaknesses, detailed comments plus:

- How does your method compares (in performances and computational cost) to traditional methods to compute segmentation length/perimeter? I.e. $s_\theta^\top L s_\theta$,
- Can you tune the "width" of the boundary under consideration, by changing the architecture of your perimeter module?
- Why did you pick a L2 norm speficically (and not L1 or something else) when minimizing the difference between perimeters?
- Can you give us your intuition on why your method works better on the selected datasets?
- Do you intend to release your code?
- Can your proposed method be used as a stand-alone loss? For instance, it has been reported that the boundary loss can work alone in multi-class settings https://github.com/LIVIAETS/boundary-loss#multi-class-setting , such as ACDC.
(I'm being cheeky, the answer is obviously no. But the paper would benefit from highlighting more the _regularizer_ aspect of the proposed loss.)
- How would your loss perform on a brain lesion dataset, where sparse (i.e. many small connected components) is a feature, not a bug? Perimeter regularization is good for single connected components, but when dealing with multiple components it might favor a single bigger one in place of the expected many small ones.






**Special Issue:**

yes

---

> ### Author Response · Authors · 2021-03-15
> **Official response to reviewer AnonReviewer2**
>
> We thank the reviewer immensely for his feedback on our proposed method.  We have modified our paper content based on their recommendations in minor changes. We have also enriched the paper with more details with regards to the hyperparameter specifications of the contour loss as recommended: “The min and max pooling layers were carried out via kernel size (3,3) and a stride of 1”.
> 1. Perimeter computation methods:
> We also thank the reviewers for drawing our attention to the multiple ways to extract the perimeter from a shape; however, with the current page limit, we are unable to integrate this information into the body of the conference paper. We will make sure to elaborate more thoroughly on this aspect when we extend the paper to a journal paper.
>
> 2. Perimeter width tuning via changing the hyper-perimeter of the contour loss:
> Indeed, the boundary length could be tuned by changing the parameters of the neural architectures of the function F. For the current publication, we have respected the initial parameters chosen by (Shit et al., 2020) to obtain the contour maps. However, we thank the reviewer for this insightful recommendation as we may explore it in future works.
>
> 3. Computational Cost:
> Given the fact that the contour extraction function is mainly composed of non-trainable layers, it does not considerably add to the complexity of the network or the computational cost. Hence, the rate of convergence boils down to the minimization of the mean squared error which is quite fast. Based on the reviewers recommendation, we have commented on the computational complexity in the body of the paper. Moreover, the reviewer has requested clarification with regards to the usage of the mean squared error, over for example the absolute error. Generally, a mean squared error function has better gradient computations than an absolute error and is naturally differentiable. That is mainly why we have chosen it for our loss function.
>
> 4. Intuition Behind why the proposed method Works:
> We believe that the proposed method works well on shapes that have varying curvatures and boundary particularities because the proposed loss is based on a contour function that can better express the variations between predicted and ground-truth contours than other representations for example distance maps.  Moreover, the proposed method minimizes directly the perimeter feature within the Unet segmentation output. Hence, it is customized to accommodate datasets where in fact the perimeter has particularities and specifications. When the loss was tested against simple shapes like circles or holes, the loss did not have an added value since the shape curvature is almost constant. In contrast, when we tested against organs like the spleen or RVC, where the organ border was characterized by curves and edges and sunken surfaces, the proposed loss was able to accommodate these particularities since it minimizes this feature directly.
>
> 5. Code Release:
> Yes the code is now released on Github via this link: https://github.com/rosanajurdi/Perimeter_loss. To ensure reproducibility, we have used the exact framework as the papers of (Kervadec et al., 2019a) and have inserted the proposed loss as well as the Hausdorf floss respectively.  The Hausdorff loss function is provided in this Github code: https://github.com/JunMa11/SegLoss/tree/master/losses_pytorch
>
> 6. Perimeter length loss as a standalone loss:
> Intuitively, no, the perimeter based loss cannot learn on its own since its main objective is to optimize the perimeter of the organ and regularize the organ boundary, as you have respectively pointed out. So, the optimization criterion or features is rather insufficient to allow proper training. However, based on the reviewer's recommendation, we are conducting experiments on the perimeter loss on its own to check the possibility of learning and will include it in the modified version of the paper.
>
> 7. Proposed loss performance datasets where small connected components is a feature:
> We thank the reviewer for his suggestion. We think it will be interesting to assess the proposed loss on multi-component shapes and organs like brain lesions and we will explore it in future work.

---

### Official Review · AnonReviewer3 · 2021-03-09

**Confidence:** 4
**Preliminary Rating:** 2
**Final Rating:** 3

**Summary:**

The paper defines a new loss term for image segmentation to optimize the perimeter of the segmented object, based on contour extraction. The loss is investigated, as a complement to Dice coefficient, on 3 public datasets of both CT and MRI images, and in both single and multi-organ segmentation contexts and proven to be superior to two reference boundary losses.

**Strengths:**

The results are given on 3 different publicly available datasets, both with CT and MRI images, with a variety of organ shapes. The proposed loss improves the segmentation performance on all datasets, and both quantitative and visual results are provided and analyzed.

**Weaknesses:**

The definition of the proposed loss is very unclear, in the text as well as in equation 2. The implementation is also questionable, since it appears to me to be a simple morphological gradient. Details should be provided on how the hyperparameters were fixed. The text is difficult to read at times.

**Deanonymize Review:**

no

**Detailed Comments:**

Boundary losses are interested in a variety of segmentation situations, including imbalanced problems. The authors argue that the average distance loss proposed by Kervadec et al. and the Hausdorff distance loss proposed by Karimi et al. fail to handle complex shapes and in particular concavities well because they are based on a point-to-point distance. I do not conquer with that analysis. Hausdorff distance is not a point-to-point distance but an actual set-to-set distance, even though the approximation proposed by Karimi et al. leverages a distance map. My opinion is rather that distances based on a distance maps tend to underestimate the contour-to-contour distances, in particular in the case of concavities, since the closest point is systematically chosen. This is an old problem in computer vision (e.g. [1] and [2]) and one of the reasons for developing the classical gradient vector flow [3] against the normal flow (as used by Boykov and Kervadec).

My main comment concerns the definition of the proposed loss which I could not get right. In the text, it is described as "the error between the sum over each respective contour via an $L_2$ norm" (p.2 introduction), "the mean squared error between the sum of the ground truth vs predicted contours" (p. 4 section 3.1), "The proposed loss then sums over the pixels belonging to organ edges for both the predicted and ground-truth regions and minimizes the $L_2$ distance between them". But these sums appear to be scalars, so I do not understand what an $L_2$ norm is used for. Isn't it just a mere squared difference between two numbers?

The same goes for equation 2. As far as I understand F computes the edge map (contour image): it is a binary image for the ground truth ($y^F_p$ = 0 or 1), and a probability map for the predicted image (gradient of the probability map produced by the U-net model, see below for the term gradient). The last (rightmost) term of equation 2 is indeed the squared difference of 2 scalars: the sum over $y^F_p$ just computes the number of pixels on the ground truth contour, hence the perimeter length; and the other sum is an approximation, of soft evaluation, of this length in the predicted image. The score seems to just be a regularization on the perimeter length (as the authors state in the last sentence of section 3.2 on p.5, except that it would be more explicit to use "perimeter length" instead of "perimeter" in that context). But it does not equal the previous term $\|F(y) - F(\hat{y})\|^2$ since $F$ produces an image. This term is equal to $\sum (\hat{y}^F_P - y^F_p)^2$ in my understanding. This is a major reason why I recommend rejection at this point: I cannot actually fathom what the proposed criterion is. Please be explicit and clear.

My second major comment is about the implementation of the contour extraction. I understand that it is just a morphological gradient or one version of it, which again is not clear.

According to Figure 2, it is the difference between the closing and erosion of the shape. This may have thin parts of object disappear. Also, I do not see what a ReLU layer is useful for since the dilation is by definition greater than the input image, which is in this case the erosion of the original image. A simple identity activation could work.

But the text reads otherwise, at the end of p. 3 (section 3.1) it appears that the gradient is computed as the difference between the dilation and the erosion of the original image. This is not the same and the generated contour may be thicker. But here again, I cannot see how the ReLU activation would be useful: the erosion ensures that the output pixel value is less than the original value, and the dilation ensures that the output pixel value is greater than the original value, hence the difference is always positive.

Again, the authors need to be more clear and consistent in their exposition. I think the term morphological gradient is enough and Figure 2 (if correct) clearly explains how it can be implemented using blocks and operations available on the GPU. ReLU activation does not seem useful (but it is harmless), and the size of the max/min pool kernels should be explicited (aka structure element).

In 4.2, it is nice that the same architectures and hyperparameters were used, though I am not sure that  boundary loss values are comparable and ponder whether the same $\lambda$ should be used. Please provide more details on how these hyperparameters were fixed: There is always a doubt that the described configuration may benefit the proposed loss. Also, why use Monte-Carlo simulation instead of 5-fold validation since you use 80/20% splits?

I think there is an inversion between the left ventricular cavity (LVC) and the myocardium (MYO) in the ACDC dataset. On  Figure 1, contrary to what the caption reads, the LVC is in yellow and the MYO is in green in my opinion. This questions all subsequent results: should MYO and LVC be inverted as well in the following tables and figures?

In eq. 4, the two $D_G$ terms should be distinguished: one is the distance to the ground truth contour and the second one is the distance to the predicted contour.

[1] Chalana V, Kim Y. A methodology for evaluation of boundary detection algorithms on medical images. IEEE Trans Med Imag. 1997; 16(5):642-52. https://doi.org/10.1109/42.640755

[2] Tagare HD. Shape-based nonrigid correspondence with application to heart motion analysis. IEEE Trans Med Imag. 1999; 18(7):570-9. https://doi.org/10.1109/42.790457

[3] Xu C and Prince JL. Snakes, shapes, and gradient vector flow. in IEEE Transactions on Image Processing, vol. 7, no. 3, pp. 359-369, March 1998, https://doi.org/10.1109/83.661186


**Final Rating Justification:**

I thank the authors who have answered my comments. I thereafter change my rating to weak accept.
A typo has been inserted in the revised text (last sentence of p. 3): back-propegation -> back-propagation

**Justification Of The Preliminary Rating:**

The proposed loss definition is unclear and makes it impossible at this point to assess its potential interest for publication. The description of the contour extraction operator should also be clarified.

**Paper Type:**

methodological development

**Questions To Address In The Rebuttal:**

- a very explicit and clear definition of the proposed loss is mandatory
- comment and/or correct the description of the contour operator
- comment on MYO/LVC potential inversion and how the results should be read.

**Special Issue:**

no

---

> ### Author Response · Authors · 2021-03-15
> **Response to reviewer 3 includes resolving clarity issues with regards to loss formulation, description, and experimental validation**
>
> We thank the reviewer for his/her constructive criticism.
> 1. Clarity and Formulation issues:
> First of all, we agree and apologize for the error in the formulation particularly with regards to the L_2 norm expression. We agree with the reviewer that the usage of L_2 norm does not conform with the concept and methodology of the proposed loss and hope that this error won't hinder their overall judgement regarding to the novelty of the method. Based on the reviewer’s clarity issue comments, we have suppressed the L_2 norm formula, adopted the perimeter length expression instead of perimeter expression, and corrected the caption in fig. 1.  We have distinguished between the distance maps of predicted and ground-truth contours in eq. 3 and 4 by the usage of yp to indicate ground-truth segmentation and \hat{yp} to indicate predicted segmentation. With regards to the contour function ,yes, it is a form of morphological gradient except that it can be applied to probability maps while still ensuring proper back-propagation. We agree this may have not been clear in the paper and have modified the text accordingly.
>
> 2. Hausdorff distance is not a point-to-point distance but an actual set-to-set distance:
> Indeed, the obvious validity of the proposed loss comes from the replacement of distance maps with contour maps extracted from both probability and ground truth maps. Nevertheless, what we meant by point-by-point optimization concerns the general (initial) definition of both boundary and Haussdorf losses.  The Hausdorff loss is a metric inspired loss, based on the Hausdorff distance. The Haussdorf distance is defined as a distance-based metric to compare contours based on point position, ie as the maximum distance between the predicted and ground-truth contour points.  Also, the initial formulation of the boundary loss is derived from the distance function which is computed point-wise meaning relative to points on the ground-truth contour and its corresponding  point on the predicted contour (ex: eq. 2 in Kervadec et al.a). After, the Loss formulation with the distance maps is obtained via Boykov’s work (Boykov et al., 2001). What we meant by point-by-point optimization is that instead of taking into consideration the distance between points on the predicted and ground-truth contours (later on transformed to distance maps formulations), we thought of taking the perimeter length between the predicted and the ground-truth contour and minimize it. We agree this may not have been clarified explicitly in the body of the paper and have modified our paper accordingly.
>
> 3. Relu Significance and Contour function Implementation : We agree with the reviewer that a simple identity function would work in our problem. However, the contour extraction function is inspired by (Shit et al.,2020, CVPR) who proposed this extraction technique given a process that would allow them to obtain the skeleton maps of vessel data. We have mentioned this in section 3.1 We have respected the implementation of the contour function as proposed originally with the ReLu function and as the difference between the erosion of the original image and the dilation of the eroded one. The output of the function when applied to the ground-truth segmentation is a contour map of values {0,1} where a pixel has a value of 1 if it is a contour pixel and 0 else. For the predicted segmentation map, the values in the contour maps is 0 for pixels not belonging to the predicted contour else, p corresponding to the probability the U-Net has given to the pixel belonging to the contour. After we obtain the contour maps of both predicted and GT segmentations, we have summed over all the pixels in the contour maps to obtain a GT perimeter length and a predicted perimeter length and have minimized the mean squared error between the two.
>
> 4. Details regarding hyperparameters and lambda: We agree that in many cases configuration may benefit a loss over the other. If we are to conduct a thorough comparison, we should finetune to obtain the optimal parameters for the proposed loss and compare relative to the boundary loss on one hand. Then, obtain the optimal parameters for the boundary loss and compare our proposed loss under these parameters. In our experiments, we have adopted the parameters and the exact same framework provided by (Kervadec et al., 2019a).  Hence this configuration is actually the optimal configuration for the competing loss: the boundary loss. Even with this framework being the optimal for the boundary loss, we still arrived at good results on all 3 datasets.
>
>
> 5. Usage of Monte-Carlo instead of K-folds: Due to computational resources and time limitations, we have evaded using 5-fold cross validation. Instead, we have chosen 3- Monte Carlo simulations which is a validation method recommended in this survey on cross-validation procedures : (Arlot and Celisse, 2010).

---

### Meta-Review · Area_Chairs · 2021-03-31

**Recommendation:** Accept (Poster)

**Metareview:**

overall good quality work with outstanding reviews and discussion. Everyone agrees this would make a nice poster presentation at MIDL.

**Paper Type:**

methodological development

---

### Decision · Program_Chairs · 2021-03-31

Accept